# Seeing is Simulating: Differentiable Physics for Interaction-Aware Material Estimation

**Chun Feng**[*]                                                     *chunf2@illinois.edu*
*University of Illinois Urbana Champaign*

**Hao Zhang**[*]                                                     *haoz19@illinois.edu*
*University of Illinois Urbana Champaign*

**Haolan Xu**[†]                                                     *jamesdemon923@gmail.com*
*University of Illinois Urbana Champaign*

**Narendra Ahuja**                                                   *n-ahuja@illinois.edu*
*University of Illinois Urbana Champaign*

[*]**Equal contribution**
[†]**Work done as a research assistant at the University of Illinois Urbana-Champaign**

**Reviewed on OpenReview:** `https://openreview.net/forum?id=lwuaTI4ISa&referrer`

## Abstract

Modeling human-object interactions is crucial for creating immersive virtual experiences. However, synthesizing 3D object dynamics conditioned on actions remains a challenging problem. Existing approaches equip static 3D objects with motion priors distilled from video diffusion models. This methodology has two drawbacks: (i) video diffusion models are not physically grounded. Thus, the generated videos may contain physical inaccuracies; (ii) video diffusion models cannot generate complex dynamics where multiple objects interact under actions with long durations and large spatial extent. We present **PhysInteract**, a physics-based framework that (i) models interactions with a representation that captures their duration and contact information; (ii) estimates object material properties (e.g., Young's modulus) from objects' deformation caused by interactions; (iii) uses physics simulation to reproduce realistic object dynamics based on estimated interactions and material properties. We highlight that PhysInteract is fully differentiable, enabling joint optimization of interaction representations and object material properties. PhysInteract achieves better performance than existing methods Zhang et al. (2024); Xie et al. (2024a). We demonstrate its superiority by quantitatively testing PhysInteract on a curated dataset. In conjunction with an additional user study, our method shows a step towards more realistic and immersive virtual experiences.

## 1 Introduction

Modeling human-object interactions is a pivotal component in creating immersive virtual experiences. It requires not only the ability to interpret interactions in the wild but also the capacity to synthesize physically plausible deformations of objects in response to these actions. However, achieving this remains a formidable challenge due to two primary factors: **(i)** Complexity of Interaction Modeling. Interactions are temporal events involving multiple entities, such as duration, contact points, and force trajectories. Accurately identifying them from unstructured video data is difficult. Furthermore, translating these abstract concepts into structured representations compatible with physics simulations poses a significant hurdle; **(ii)**

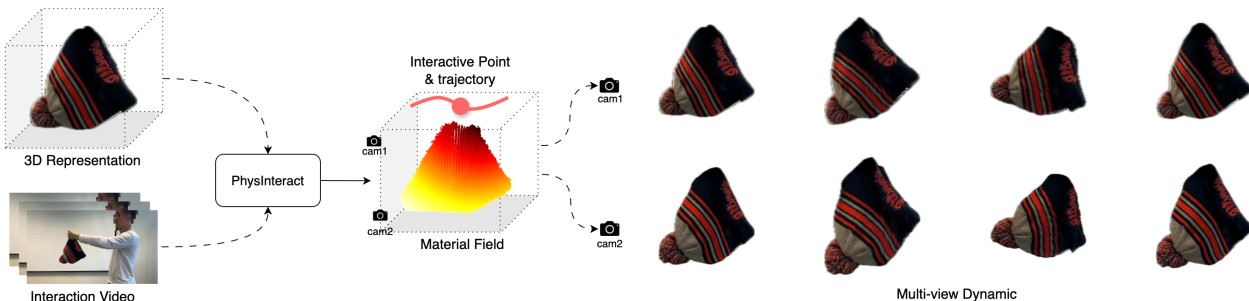

Figure 1: PhysInteract is designed to model human-object dynamics. Given an object's 3D representation and a video depicting interactions with the object, it first models interactions using the trajectories of a set of interactive points, then estimates the object's material properties (e.g., Young's modulus), and finally uses physics simulation to reproduce the realistic object dynamics shown in the input video.

Requirement for Causal Physical Understanding. Entities in an interaction undergo continuous state changes. For instance, velocities and poses are altered by external forces, and non-rigid objects deform upon contact. Capturing these causal relationships is essential for the accurate modeling of human-object interactions, yet it requires reasoning beyond simple visual correspondence.

Prior work has predominantly used statistical methods to address these challenges. Leveraging the generative power of diffusion models, several approaches Chen et al. (2024b); Li et al. (2024); Zhang et al. (2023); Ren et al. (2023); Zhao et al. (2023) animate still images or 3D assets based on user inputs, such as text prompts or direct manipulation (e.g., dragging). While these methods yield visually compelling results, they are generally not physically grounded. Consequently, artifacts and physically impossible distortions are frequently observed. To mitigate this, recent research has incorporated explicit physics priors. Methods like Xie et al. (2024a); Zhang et al. (2024) utilize the Material Point Method (MPM) Jiang et al. (2016) to simulate 3D dynamics based on material properties derived from video motion priors. However, these approaches often rely on an oversimplified assumption where object motion is driven by inertia from an initial velocity. They lack a mechanism to model continuous, external human-object interactions, rendering them incapable of handling complex dynamics in which objects are subjected to sustained actions over long durations and extensive spatial contact.

To bridge this gap, we propose *PhysInteract*, a fully differentiable, physics-based framework designed to accurately model human-object interactions. Given a 3D representation (e.g., Gaussian Splatting Kerbl et al. (2023) or mesh) and a monocular video depicting the interaction as input, PhysInteract operates through three integrated stages: **(i)** Interaction Representation Construction. Instead of relying on simple initial states, PhysInteract explicitly models the interaction trajectory. By leveraging vision-language models and segmentation tools (e.g., SAM Ravi et al. (2024)) to localize contact areas, we back-project these regions into 3D space as "contact points". We then track these "contact points" over time to construct a coarse interaction trajectory that drives the object's deformation; **(ii)** Differentiable Physics Simulation. We employ a differentiable MPM simulator to model 3D dynamics. This simulator requires intrinsic material properties, such as Young's modulus (stiffness) and Poisson's ratio (deformability), which are initially unknown. PhysInteract initializes these parameters randomly or via heuristics; **(iii)** Joint Optimization. A core innovation of PhysInteract is the joint optimization of interaction representations and material properties. Since the initially extracted trajectories may be inaccurate, simply optimizing material properties leads to suboptimal solutions. Therefore, PhysInteract iteratively refines both the contact point trajectories (forces) and the object's material parameters by minimizing the discrepancy between the rendered simulation and the original video frames. This "analysis-by-synthesis" loop allows us to recover accurate physical parameters and realistic motion simultaneously.

To evaluate our method, we curate a dataset featuring objects with diverse material properties (e.g., elastic rubber, jelly) undergoing various interactions (e.g., pulling, dragging). Quantitative experiments demonstrate that PhysInteract significantly outperforms existing approaches in modeling fidelity. Furthermore,

extensive user studies validate our method's superiority in motion realism and visual quality. Notably, Phys-Interact exhibits strong generalization capabilities: once material properties are estimated, the system can generate realistic object dynamics conditioned on novel, unseen interactions without retraining.

To summarize, our contributions are threefold:

- We propose a novel framework, PhysInteract, that accurately models human-object interactions by jointly estimating interaction trajectories and material properties within a unified, differentiable optimization process.

- We introduce a comprehensive dataset containing diverse human-object interaction samples to facilitate future research in this domain.

- We conduct extensive experiments and user studies, demonstrating state-of-the-art visual quality and the ability to synthesize dynamics conditioned on novel interactions.

## 2 Related Work

**4D content generation.** This field aims to synthesize 4D representations that maintain both multi-view and temporal consistency. A prevalent paradigm involves integrating 3D generation pipelines with video diffusion models. For instance, Xie et al. (2024b); Bahmani et al. (2024); Yin et al. (2023); Ling et al. (2024); Zhao et al. (2023); Jiang et al. (2023); Chen et al. (2024a); Ren et al. (2023); Singer et al. (2023); Zeng et al. (2024); Yang et al. (2024) extend static 3D representations to the temporal dimension via Score Distillation Sampling loss Poole et al. (2022). While yielding plausible visual results, these methods inherently struggle with the high computational cost and instability associated with SDS and video diffusion models. Furthermore, due to the hallucination problem characteristic of generative models Radford et al. (2018), they often fail to ensure accurate multi-view consistency or kinematic plausibility. Alternatively, geometry-based methods Pan et al. (2024); Sun et al. (2024); Kratimenos et al. (2024); Duan et al. (2024); Das et al. (2024); Liang et al. (2025); Lin et al. (2024) adapt 3D reconstruction pipelines to 4D content by utilizing multi-view images at every timestep. Although these approaches offer superior optimization efficiency, their applicability is severely constrained by the scarcity of high-quality, synchronized multi-view 4D data, limiting their generalization potential.

In contrast, PhysInteract generates plausible 4D content using only monocular videos. By integrating a forward-pass simulation engine, we ensure strict physicality and temporal consistency without relying on computationally intensive generative priors.

**Simulation-based generation.** To enhance physical realism, recent works Feng et al. (2024); Li et al. (2023); Xie et al. (2024a) impart physical properties to 3D representations, endowing them with kinematic attributes (e.g., velocity, strain) and mechanical properties (e.g., stiffness, plasticity). These representations are then driven by the laws of continuum mechanics to produce 4D dynamics. A central challenge in this domain is obtaining the accurate physical parameters required for simulation. Early approaches Tan et al. (2024); Xie et al. (2024a); Zhao et al. (2024); Lin et al. (2025) derive such knowledge from human priors or Large Language Models (LLMs). However, these methods often suffer from a domain gap between semantic descriptions and precise physical coefficients, resulting in inaccuracies. Consequently, recent efforts have shifted towards optimization-based parameter estimation. For example, Zhang et al. (2024); Huang et al. (2025); Liu et al. (2024a) distill dynamic priors from video generation models. However, as discussed in Section 1, these methods predominantly focus on inertial motion initiated by an initial state, lacking the capability to model complex scenarios involving continuous, external human-object interactions. Moreover, relying on generative video priors can introduce accumulated errors in physical parameter estimation.

In this work, we address these limitations by learning directly from real-world interactions. PhysInteract jointly optimizes interaction forces and material properties, ensuring that the synthesized dynamics are grounded in accurate, real-world physical observations.

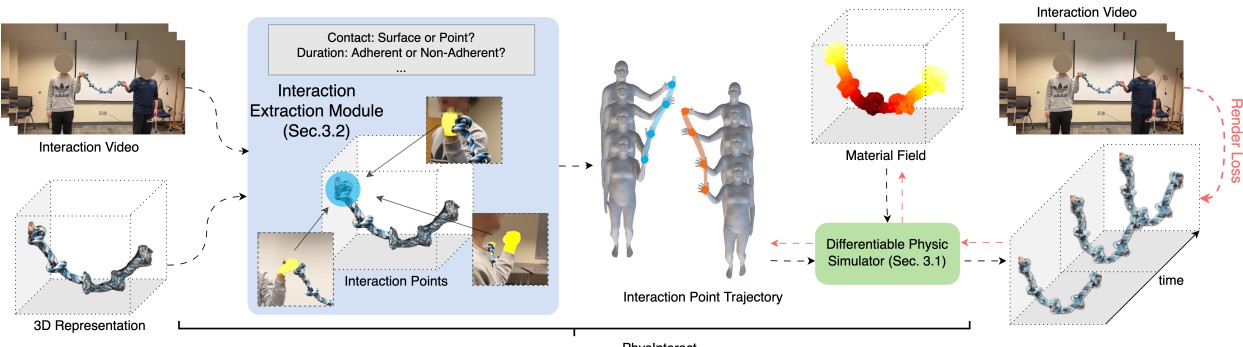

Figure 2: Overview of PhysInteract. Given an interaction video and a static 3D representation, our framework jointly estimates the physical material properties and interaction dynamics. The Interaction Extraction Module (Section 3.2) first leverages vision-language models to infer interaction semantics (e.g., contact type) and employs grounding models (Segment Anything Model Ravi et al. (2024)) to localize 3D contact points. This initializes a coarse Interaction Point Trajectory. This trajectory, along with a learnable Material Field (representing per-particle Young's modulus), is fed into a Differentiable Physics Simulator (Section 3.1). Through an analysis-by-synthesis loop, we compute a Render Loss between the simulated 4D dynamics and the observed video frames. As indicated by the red dashed arrows, gradients are back-propagated to jointly optimize both the material field and the interaction trajectory (Section 3.3), ensuring physically plausible motion reconstruction.

## 3 Method

**Overview.** Given a monocular video capturing an object undergoing interaction-driven motion, along with its static 3D representation (e.g., 3D Gaussians Kerbl et al. (2023) or mesh), our goal is to jointly estimate the object's physical material properties (e.g., Young's modulus) and the precise interaction dynamics (e.g., contact forces/trajectories). As illustrated in Figure 2, PhysInteract operates in three stages: **(1)** Initialization. We first leverage vision-language and tracking models to construct a coarse representation of the interaction (Section 3.2); **(2)** Joint Optimization. We employ a differentiable physics simulator to iteratively refine both the material properties and the interaction trajectory by minimizing the discrepancy between simulated renderings and observed video frames (Section 3.3); **(3)** Inference. Finally, the learned physical model enables the synthesis of realistic dynamics conditioned on novel interactions (Section 3.4).

### 3.1 Preliminaries

**3D representation.** We adopt 3D Gaussian Splatting (3DGS) Kerbl et al. (2023) as our primary representation due to its explicit Lagrangian nature, which is naturally compatible with particle-based physics simulators. The object is parameterized as a set of Gaussians $\{\mathcal{G}_p\}_{p=1}^{P}$, where each particle $\mathcal{G}_p$ possesses position $\boldsymbol{x}_p$, covariance $\Sigma_p$, opacity $\alpha_p$, and color $\boldsymbol{c}_p$. While we focus on 3DGS, our framework is agnostic and also compatible with mesh-based representations.

**Differentiable material point method.** To simulate physically grounded motion, we utilize the Material Point Method (MPM) Jiang et al. (2016), a hybrid Lagrangian-Eulerian solver. The simulation follows the laws of continuum mechanics, governed by the conservation of mass and momentum. The object's deformation is characterized by the deformation gradient $\boldsymbol{F}$, and the internal stress $\sigma$ is computed via a constitutive model. We employ the Fixed Corotated elasticity model, where the stress-strain relationship is defined by the energy density function $\psi(\boldsymbol{F})$:

$$\psi(\boldsymbol{F}) = \mu \sum_{i=1}^{d} (\sigma_i - 1)^2 + \frac{\lambda}{2}(\det(\boldsymbol{F}) - 1)^2, \tag{1}$$

Here, $\sigma_i$ are the singular values of $\boldsymbol{F}$. The Lamé parameters $\mu$ and $\lambda$ are derived from the object's material properties: Young's modulus $E$ (stiffness) and Poisson's ratio $\nu$ (compressibility). Crucially, our MPM simulator is *fully differentiable.* This allows gradients to propagate from the rendered image loss back through the simulation steps to update both the initial material parameters $(E, \nu)$ and the external interaction forces.

### 3.2 Initialization: Interaction Representation

Directly optimizing physics parameters from scratch is prone to local minima. Therefore, we propose a multi-stage initialization strategy to construct a coarse but semantically meaningful interaction representation.

**Semantic parsing via vision-language models.** We first employ a Vision-Language Model (VLM) Bai et al. (2023) to parse the video context. The VLM identifies: (i) interaction type (e.g., point-based contact vs. surface collision); and (ii) temporal characteristics (e.g., continuous grasping vs. instantaneous impact). This semantic prior guides the subsequent geometric extraction.

**Contact point localization.** To locate the physical contact, we utilize an open-vocabulary grounding model Liu et al. (2024b) to detect the interaction region (e.g., "hand") bounding box $\mathbf{B}$, followed by the Segment Anything Model (SAM) Ravi et al. (2024) to extract a precise 2D contact mask $M_{\text{contact}}$. To lift this 2D information into 3D, we estimate the camera pose $\mathbf{T}_c$ and object pose $\mathbf{T}_o$ via differentiable rendering alignment (optimizing RGB and silhouette consistency). We then back-project the masked pixels $\mathbf{x}_p \in M_{\text{contact}}$ into 3D space using the estimated depth $d_p$:

$$\mathbf{X}_p = (\mathbf{T}_c)^{-1}\pi^{-1}(\mathbf{x}_p, d_p). \tag{2}$$

This yields a set of 3D particles $\mathcal{P}_{\text{contact}}$ representing the interaction interface. This process is similar to that described in Yang et al. (2021).

**Trajectory initialization.** An interaction is defined not just by contact, but by motion. We initialize the interaction trajectory $\mathcal{T}^{(0)}$ based on the scenario: for human-object interaction, we lift 2D tracks from a human pose tracker (e.g., 4D Humans Goel et al. (2023)) to 3D; for object-object interaction, we utilize scene flow tracking Wang et al. (2025). This initial trajectory $\mathcal{T}^{(0)}$ serves as a "soft guide" for the subsequent physics-based optimization.

### 3.3 Physics-Aware Joint Optimization

The initialization described above relies on visual observations, which effectively capture kinematics but lack physical causality. Therefore, we propose a joint optimization framework that refines both the interaction dynamics and the material properties through an analysis-by-synthesis loop.

**Parameterization.** We optimize two sets of parameters:

- **Material Field** $\Theta_{\text{mat}}$: We parameterize the physical properties explicitly on the particles. Specifically, we assign a Young's modulus $E_p$ and a Poisson's ratio $\nu_p$ to each particle $p$ (including both surface Gaussians and internal filling particles). This enables the representation of spatially varying material properties that naturally deform with object geometry.

- **Interaction Trajectory** $\mathcal{T}$: We parameterize the interaction as a sequence of velocity boundary conditions or external forces $\{\mathbf{f}_t\}_{t=1}^T$ applied to the contact particles $\mathcal{P}_{\text{contact}}$.

**Forward simulation and rendering.** At each optimization step, the simulator takes the current material field $\Theta_{\text{mat}}$ and interaction $\mathcal{T}$ to simulate the object's dynamics from $t = 0$ to $t = T$, producing a sequence of deformed particle states $\{\mathbf{x}_t\}_{t=1}^T$. These states are then rendered into images $\{\hat{I}_t\}_{t=1}^T$ via a differentiable Gaussian splatting renderer proposed in Kerbl et al. (2023).

Table 1: Dataset details. We evaluate PhysInteract on 11 scenes comprising diverse object types (Synthetic and Real) and multiple interaction modalities.

| Scene | Source | Object | Interaction Type |
|-------|--------|--------|------------------|
| 1 | Sync | Hemp rope | Point Adherent |
| 2 | Sync | Bow tie | Point Adherent |
| 3 | Sync | Tow rope | Point Adherent |
| 4 | Sync | Glove | Point Adherent |
| 5 | Sync | Balloon Dog | Point Adherent |
| 6 | Sync | Basketball | Surface Non-Adherent |
| 7 | Sync | Balloon | Point Adherent |
| 8 | Sync | Flower | Point Adherent |
| 9 | Real | Knit cap | Point Adherent |
| 10 | Real | Telephone cable | Point Non-Adherent |
| 11 | Real | Toy rope | Point Adherent |

**Objective function.** We optimize $\Theta_{\mathrm{mat}}$ and $\mathcal{T}$ jointly by minimizing a composite loss function:

$$\mathcal{L}_{\mathrm{total}} = \mathcal{L}_{\mathrm{rec}} + \lambda_{\mathrm{reg}} \cdot \mathcal{L}_{\mathrm{reg}}. \tag{3}$$

The reconstruction loss $\mathcal{L}_{\mathrm{rec}}$ enforces visual fidelity:

$$\mathcal{L}_{\mathrm{rec}} = \sum_t \left( \|\hat{I}_t - I_t\|_1 + \lambda_{\mathrm{ssim}}(1 - \mathrm{SSIM}(\hat{I}_t, I_t)) + \lambda_{\mathrm{mask}}\|\hat{M}_t - M_t\|_2 \right), \tag{4}$$

where $I_t, M_t$ are ground-truth video frames and silhouettes. To ensure physical plausibility and prevent overfitting to visual noise, we apply regularization $\mathcal{L}_{\mathrm{reg}}$:

$$\mathcal{L}_{\mathrm{reg}} = \|\nabla\Theta_{\mathrm{mat}}\|_2 + \|\mathcal{T} - \mathcal{T}^{(0)}\|_2, \tag{5}$$

which encourages spatial smoothness in the material field and constrains the optimized trajectory to remain semantically consistent with the visual initialization $\mathcal{T}^{(0)}$.

### 3.4 Inference: Novel Interaction Synthesis

Once the optimization converges, we obtain a physically calibrated representation of the object, encapsulated in the estimated material field $\Theta_{\mathrm{mat}}^*$, which allows PhysInteract to generalize. We can now discard the original interaction $\mathcal{T}$ and apply novel, user-defined interactions $\mathcal{T}_{\mathrm{new}}$ (e.g., applying a new force at a different location). By running the forward MPM simulation with $\Theta_{\mathrm{mat}}^*$ and $\mathcal{T}_{\mathrm{new}}$, we synthesize realistic 4D dynamics that are not present in the original video, enabling applications such as interactive virtual editing.

### 3.5 Implementation Details

**Internal filling.** Standard 3DGS representations only model the object surface, whereas physics simulations require a continuum volume. To address this, we implement an internal filling strategy. For mesh inputs, we perform voxelization. For Gaussian inputs, we first densify particles within the Gaussian ellipsoids to approximate local volume, and then apply voxel-based filling to populate the interior. These internal particles are assigned the same material properties as their nearest surface neighbors and participate in the MPM simulation to ensure structural stability.

**Training details.** We implement PhysInteract using PyTorch and the Taichi programming language Hu (2018) for differentiable MPM simulation. The optimization is performed on a single NVIDIA A40 GPU. We use the AdamW optimizer Loshchilov & Hutter (2017) for both material properties and interaction trajectories. The learning rates are initialized at $1 \times 10^{-4}$ for material parameters and $5 \times 10^{-5}$ for trajectory updates, with a cosine annealing schedule. The optimization typically converges within $5,000$ iterations, taking approximately 30 minutes per video sequence.

Table 2: Quantitative comparison. We report SSIM ($S$) and PSNR ($P$) across all 11 scenes. Baselines include PhysDreamer (PD), DreamGaussian4D (DG4D), and GPT+PhysGaussian (GPT-PG). Our method consistently achieves state-of-the-art reconstruction quality.

| Method | Scene 1 | | Scene 2 | | Scene 3 | | Scene 4 | | Scene 5 | | Scene 6 | |
|--------|------|-------|------|-------|------|-------|------|-------|------|-------|------|-------|
| | $S$ | $P$ | $S$ | $P$ | $S$ | $P$ | $S$ | $P$ | $S$ | $P$ | $S$ | $P$ |
| PD | 0.63 | 14.78 | 0.62 | 15.05 | 0.67 | 17.23 | 0.53 | 14.62 | 0.61 | 14.03 | 0.52 | 13.99 |
| DG4D | 0.63 | 14.18 | 0.64 | 15.32 | 0.71 | 17.89 | 0.64 | 14.81 | 0.63 | 14.63 | 0.62 | 14.52 |
| GPT-PG | 0.66 | 15.05 | 0.68 | 17.23 | 0.42 | 12.02 | 0.58 | 15.88 | 0.62 | 14.01 | 0.66 | 15.98 |
| Ours | **0.89** | **22.38** | **0.87** | **21.37** | **0.81** | **20.75** | **0.88** | **23.71** | **0.91** | **21.94** | **0.89** | **22.56** |

| Method | Scene 7 | | Scene 8 | | Scene 9 | | Scene 10 | | Scene 11 | | Average | |
|--------|------|-------|------|-------|------|-------|------|-------|------|-------|------|-------|
| | $S$ | $P$ | $S$ | $P$ | $S$ | $P$ | $S$ | $P$ | $S$ | $P$ | $S$ | $P$ |
| PD | 0.67 | 16.81 | 0.61 | 15.48 | 0.46 | 14.79 | 0.49 | 9.33 | 0.64 | 12.70 | 0.58 | 14.44 |
| DG4D | 0.64 | 15.79 | 0.62 | 14.21 | 0.46 | 13.80 | 0.54 | 9.96 | 0.51 | 10.87 | 0.60 | 14.18 |
| GPT-PG | 0.68 | 16.01 | 0.61 | 15.89 | 0.51 | 14.21 | 0.50 | 13.34 | 0.57 | 11.01 | 0.59 | 14.60 |
| Ours | **0.75** | **20.63** | **0.94** | **23.94** | **0.73** | **20.65** | **0.71** | **19.5** | **0.94** | **24.92** | **0.84** | **22.03** |

## 4 Experiments

### 4.1 Experimental setup

**Dataset.** Due to the scarcity of high-quality 4D interaction data with ground truth physical properties, we curated a hybrid dataset comprising both synthetic and real-world scenes. As detailed in Table 1, our dataset consists of 11 diverse scenes covering: **(1)** Synthetic Scenes (8 sequences): we simulated objects with varying geometries (e.g., hemp rope, bow tie, balloon dog) and material properties using accurate physics engines, which provides ground truth for quantitative physical evaluation. **(2)** Real-world Scenes (3 sequences): we captured real-world interactions involving a telephone cable, a knit cap, and a toy rope. Crucially, the dataset spans multiple interaction types, including "Point Adherent" (e.g., grasping), "Point Non-Adherent" (e.g., poking), and "Surface Non-Adherent" (e.g., bouncing), ensuring comprehensive robustness evaluation.

**Implementation details.** We leverage a vision-language model Bai et al. (2023) to analyze visual content and infer interaction types by framing the prompt as a multiple-choice question. Following this, we employ SAM Ravi et al. (2024) to segment interaction regions. For human-object scenarios, 4D-Humans Goel et al. (2023) is used to initialize trajectories; otherwise, we utilize scene flow tracking Wang et al. (2025). To ensure computational efficiency, we crop foreground objects into a normalized $[0.0, 3.0]^3$ space discretized by a $100^3$ MPM grid. Objects are represented by approximately $5,000$ surface particles. Moreover, to guarantee simulation stability, we augment these with $5,000$ to $10,000$ internal filling particles, which share the same physical properties as surface particles but are excluded from rendering.

**Baselines.** We compare PhysInteract against three state-of-the-art approaches. We analyze their mechanisms and limitations below:

- **PhysDreamer** Zhang et al. (2024): estimates physical properties using video priors from diffusion models. *Limitation:* It models motion via a velocity field that only captures internal deformation. Consequently, it fails to account for external forces in interaction scenarios, leading to incorrect velocity predictions.

- **DreamGaussian4D** Ren et al. (2023): optimizes 4D Gaussians directly from video via deformation fields. *Limitation:* It lacks explicit physical modeling. As particle counts increase, the optimization landscape becomes highly complex, causing it to struggle with simultaneous large deformations and external interactions.

- **GPT + PhysGaussian** Xie et al. (2024a): adapts PhysGaussian by using GPT Achiam et al. (2023) to infer material parameters from video, which are then fed into the simulator. *Limitation:* LLMs often struggle to map visual observations to precise numerical physical coefficients (e.g., Young's modulus), leading to plausible but physically inaccurate simulations.

Table 3: Material estimation accuracy on synthetic scenes. We compare our full model against a variant without joint trajectory optimization (*w/o JO*). The results show that refining trajectories is essential for accurate material estimation.

| Method | Scene 1 | Scene 2 | Scene 3 | Scene 4 | Scene 5 | Scene 6 | Scene 7 | Scene 8 | Average |
|---|---|---|---|---|---|---|---|---|---|
| w/o JO | 0.721 | 0.685 | 0.654 | 0.782 | 0.710 | 0.745 | 0.690 | 0.805 | 0.724 |
| Ours | **0.853** | **0.812** | **0.794** | **0.887** | **0.823** | **0.861** | **0.804** | **0.910** | **0.843** |

## 4.2 Quantitative Evaluation

**Visual fidelity.** We first evaluate the visual quality of the reconstructed motion using the Structural Similarity Index (SSIM) Wang et al. (2004) and the Peak Signal-to-Noise Ratio (PSNR). As shown in Table 2, PhysInteract consistently outperforms all baselines across all 11 scenes. Specifically, we achieve an average PSNR of 22.03, significantly surpassing the best baseline (GPT+PhysGaussian: 14.60). This indicates that our joint optimization of physics and interaction trajectories yields the most faithful reconstruction of observed dynamics.

**Physical Parameter Estimation.** We further evaluate the accuracy of estimated physical properties on our synthetic dataset, where the ground-truth Young's modulus $E^{\text{gt}}$ is available. We define the Material Accuracy (MA) metric as the percentage of particles where the estimated modulus $E^{\text{pred}}$ falls within a tolerance threshold $\delta$ (set to 0.2):

$$\text{MA} = \frac{1}{P} \sum_{p=1}^{P} \mathbb{I} \left( \frac{|E_p^{\text{pred}} - E_p^{\text{gt}}|}{E_p^{\text{gt}}} \leq \delta \right) \tag{6}$$

The results are reported in Table 3. PhysInteract (Ours) achieves a high average accuracy of 84.3%, demonstrating that our method can reliably recover underlying physical properties from monocular video.

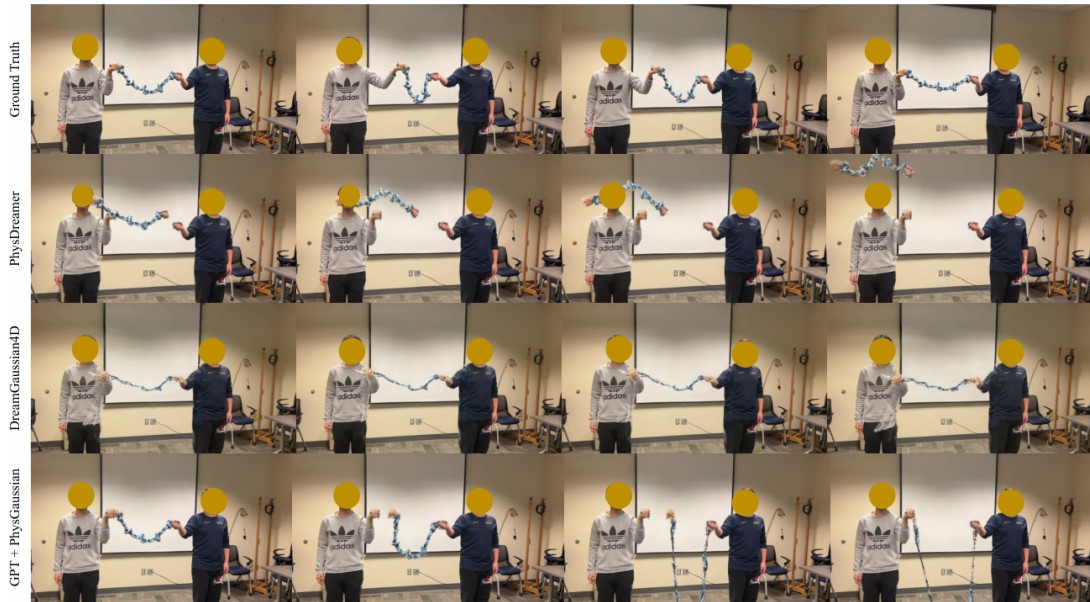

Figure 3: Visual comparison against baseline methods. We compare PhysInteract with PhysDreamer Zhang et al. (2024), DreamGaussian4D Ren et al. (2023), and GPT+PhysGaussian Xie et al. (2024a) on a real-world "toy rope" sequence. PhysDreamer generates distorted geometries due to the lack of explicit interaction modeling. DreamGaussian4D exhibits artifacts and a loss of structural coherence during large motions. GPT+PhysGaussian produces plausible but inaccurate motion dynamics due to incorrect material estimation. In contrast, PhysInteract (Ours) faithfully reconstructs the interaction dynamics, closely matching the Ground Truth.

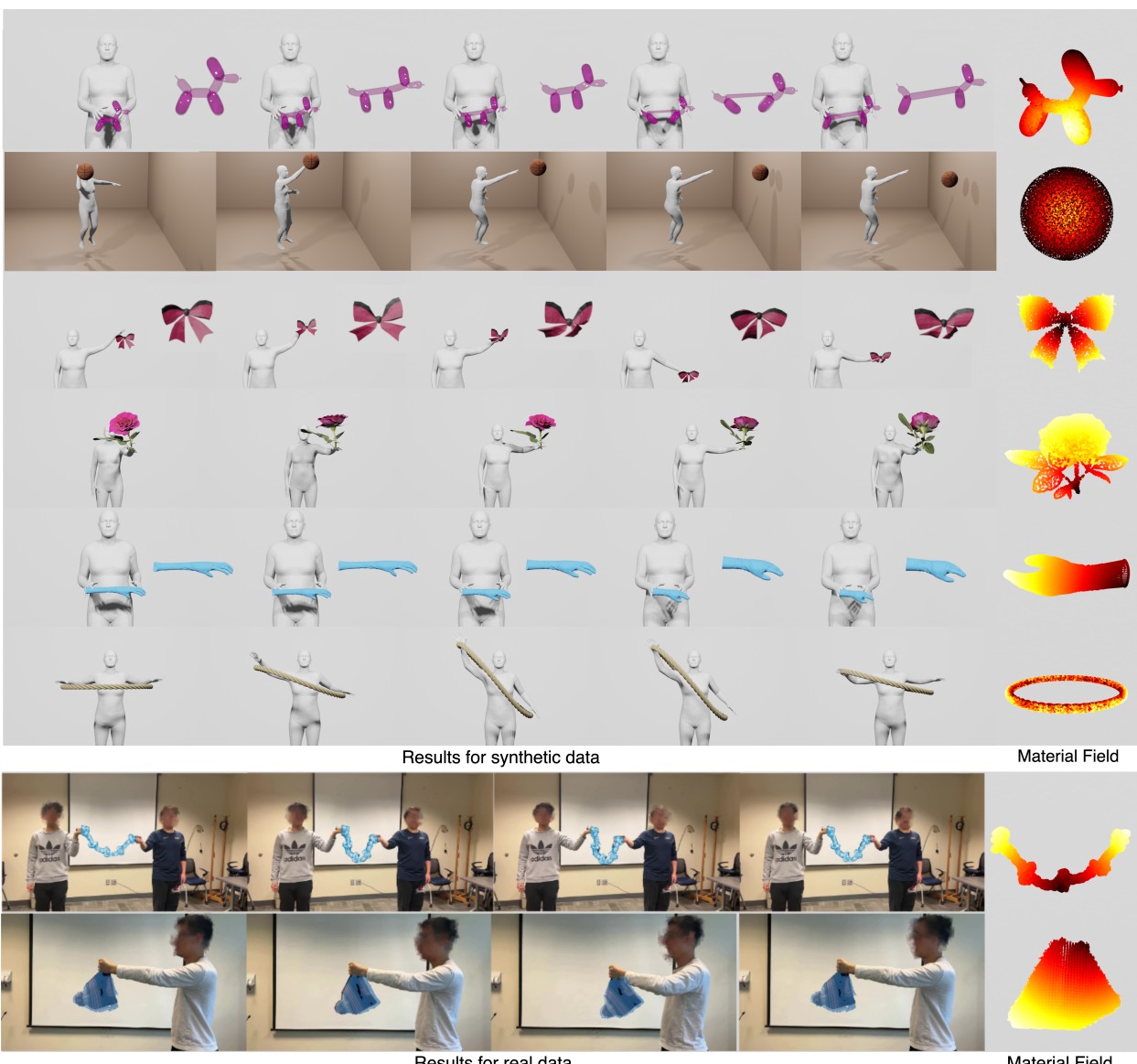

Figure 4: Qualitative results of PhysInteract. We visualize the reconstructed interaction sequences across diverse scenarios, including synthetic data (top three rows) and real-world captures (bottom two rows). For each sequence, we display key frames of the interaction. The rightmost column visualizes the learned material field (Young's modulus), where warmer colors indicate higher stiffness. Our method successfully reconstructs complex deformations (e.g., balloon twisting) and interactions (e.g., rope pulling) while recovering spatially consistent physical properties.

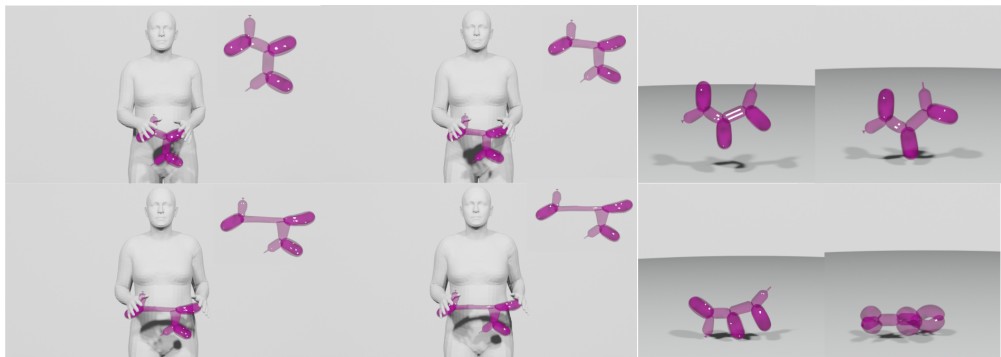

Figure 5: Generalization to novel interactions. Once the material field is learned from a reference video, PhysInteract enables *zero-shot* simulation of unseen interactions. Here, we apply two distinct new forces to the balloon dog (Top: pulling the leg; Bottom: compressing the body). The object responds with physically plausible elastic deformations, demonstrating that our method learns intrinsic physical properties rather than overfitting to the training video.

### 4.3 Qualitative Analysis

**Comparison with Baselines.** Figure 3 and Figure 4 present visual comparisons. Baseline methods struggle to capture the causal relationship between interaction and deformation. For instance, in Scene 5 (Balloon Dog), baselines fail to model the elastic rebound caused by the point interaction, whereas PhysInteract faithfully reconstructs the physical response.

**Generalization to Novel Interactions.** A key advantage of estimating intrinsic physical properties is the ability to generalize. As shown in Figure 5, once the material field is learned, PhysInteract can synthesize physically plausible dynamics under *novel, unseen interactions* (e.g., applying forces to different parts of the rope) without any re-training. This capability is unique to our physics-based formulation.

Table 4: User study results. Human ratings on Motion Realism (MR) and Visual Quality (VQ). Abbreviations: PD (PhysDreamer), DG4D (DreamGaussian4D), GPT-PG (GPT+PhysGaussian).

| Method | Scene 1 | | Scene 2 | | Scene 3 | | Scene 4 | | Scene 5 | | Scene 6 | |
| --- | --- | --- | --- | --- | --- | --- | --- | --- | --- | --- | --- | --- |
| | *MR* | *VQ* | *MR* | *VQ* | *MR* | *VQ* | *MR* | *VQ* | *MR* | *VQ* | *MR* | *VQ* |
| PD | 1.8 | 2.0 | 1.5 | 1.7 | 2.2 | 2.5 | 1.9 | 2.1 | 2.0 | 2.3 | 1.6 | 1.9 |
| DG4D | 3.8 | 4.0 | 3.5 | 3.7 | 4.0 | 4.2 | 3.9 | 4.1 | 3.4 | 3.2 | 3.3 | 3.2 |
| GPT-PG | 3.9 | 4.1 | 3.6 | 3.8 | 4.1 | 4.3 | 3.8 | 4.0 | 3.1 | 2.9 | 3.2 | 2.8 |
| Ours | **4.5** | **4.7** | **4.3** | **4.6** | **4.6** | **4.8** | **4.4** | **4.7** | **4.2** | **4.5** | **4.3** | **4.6** |
| Method | Scene 7 | | Scene 8 | | Scene 9 | | Scene 10 | | Scene 11 | | Average | |
| | *MR* | *VQ* | *MR* | *VQ* | *MR* | *VQ* | *MR* | *VQ* | *MR* | *VQ* | *MR* | *VQ* |
| PD | 1.7 | 2.0 | 1.6 | 1.9 | 1.4 | 1.7 | 1.5 | 1.8 | 1.3 | 1.6 | 1.7 | 2.0 |
| DG4D | 3.9 | 4.1 | 3.7 | 3.9 | 3.4 | 3.6 | 3.5 | 3.7 | 3.3 | 3.5 | 3.6 | 3.8 |
| GPT-PG | 3.8 | 4.0 | 3.5 | 3.7 | 3.3 | 3.5 | 3.4 | 3.6 | 3.2 | 3.4 | 3.5 | 3.7 |
| Ours | **4.4** | **4.6** | **4.2** | **4.5** | **4.0** | **4.3** | **4.1** | **4.4** | **3.9** | **4.2** | **4.3** | **4.5** |

### 4.4 User Study

We conducted a user study with 50 participants to evaluate *Motion Realism (MR)* and *Visual Quality (VQ)* on a 5-point Likert scale. Methods were anonymized and presented in random order. Table 4 summarizes the results. PhysInteract significantly outperforms baselines, achieving an average MR of 4.3, compared to the second-best (DreamGaussian4D) at 3.6. This confirms that our physics-grounded approach aligns better with human perception of realistic interaction.

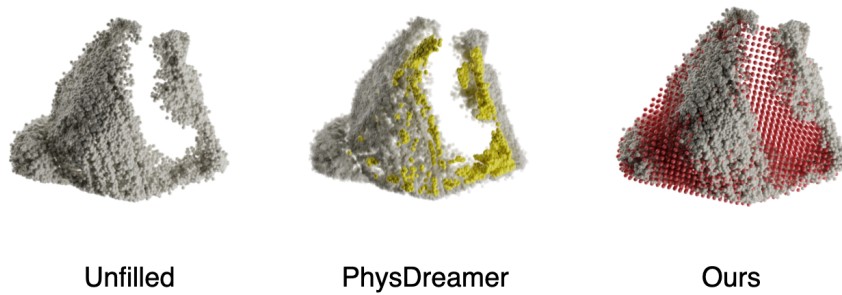

Figure 6: Ablation on internal filling strategy. We compare particle distributions for the simulation. (Left) Raw 3D Gaussians are too sparse for continuum mechanics. (Middle) PhysDreamer's mesh-based voxelization fails to capture the volume, resulting in structural holes (highlighted in yellow) that destabilize the simulation. (Right) Our approach, which densifies particles within Gaussian ellipsoids, produces a robust and dense continuum volume (red particles), ensuring stable and accurate physics simulation.

## 4.5  Ablation Study

**Impact of joint optimization.**  A core contribution of PhysInteract is the joint optimization of both material properties and interaction trajectories. To validate this, we compare our full model against a variant (w/o JO) in which the interaction trajectory is fixed to the tracker's initialization and only material properties are optimized. As shown in the second row of Table 3, excluding trajectory optimization leads to a significant drop in material estimation accuracy (Avg. decreases from 0.843 to 0.715). This is because initial trajectories from vision trackers are often noisy; without refinement, the optimizer forces the material parameters to overfit these noisy trajectories, resulting in incorrect physical properties.

**Effectiveness of Internal Filling.**  We validate our internal filling strategy by comparing it against the voxelization-based filling used in PhysDreamer. As visualized in Figure 6, converting sparse 3D Gaussians directly to a mesh for voxelization often results in holes and artifacts. In contrast, our approachdensifying particles within Gaussian ellipsoids before filling, produces a robust continuum representation, ensuring stable MPM simulation.

## 5  Conclusion and Limitations

**Conclusion.**  In this work, we introduced PhysInteract, a unified framework for modeling physically plausible human-object interactions from monocular videos. By bridging the gap between 3D visual representations and differentiable physics simulation, PhysInteract addresses the critical challenge of inferring causal physical dynamics from purely visual observations. Our core innovation lies in the joint optimization strategy, which iteratively refines both the object's material properties and the interaction trajectories within a differentiable MPM solver. Extensive experiments on a curated dataset of synthetic and real-world scenarios demonstrate that PhysInteract significantly outperforms existing diffusion-based and kinematic approaches in terms of visual fidelity and motion realism. Furthermore, by recovering intrinsic physical parameters, our method enables zero-shot synthesis of dynamics under novel user-defined interactions, taking a significant step towards creating interactive and immersive virtual experiences.

**Limitations.**  Despite its promising results, PhysInteract has several limitations that suggest avenues for future research. **(1)** Computational Complexity: while differentiable MPM ensures physical accuracy, it is computationally intensive compared to purely kinematic or generative methods. Optimizing a single sequence currently requires approximately 30 minutes, limiting its application in real-time scenarios. Future work could explore neural surrogates or reduced-order models to accelerate the simulation loop. **(2)** Dependence on 3D Initialization: our method relies on the quality of the initial static 3D representation (e.g., 3DGS). Severe occlusions or reconstruction artifacts in the initial 3D model can propagate errors into the physical simulation. Integrating dynamic 3D reconstruction jointly with physics optimization could mitigate this

issue. **(3)** Material Complexity: we model objects using a hyperelastic constitutive model suitable for a wide range of solids. However, modeling extremely complex materials (e.g., fluids, fracturing objects, or cloth with intricate weaving) remains a challenge. Extending our differentiable pipeline to support a broader library of constitutive models is a promising direction for future exploration.

## Acknowledgements

The support of the Office of Naval Research under grant N00014-20-1-2444 and of USDA National Institute of Food and Agriculture under grant 2020-67021-32799/1024178 are gratefully acknowledged. This research used the Delta system at NCSA through allocation CIS240124 from the ACCESS program, which is supported by National Science Foundation grants #2138259, #2138286, #2138307, #2137603, and #2138296.

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

# A   Appendix

## A.1   Challenges in Data Curation for Differentiable Physics

We introduce a dataset containing diverse human-object interaction samples. While we acknowledge its limited size (11 examples), we wish to clarify that this scale is not merely due to annotation constraints, but stems from the **strict physical and geometric requirements** necessary for convergent and meaningful differentiable physics optimization. Unlike purely kinematic methods, PhysInteract relies on an "Analysis-by-Synthesis" loop where gradients are back-propagated from pixel differences to physical parameters. This mechanism imposes stringent constraints on data selection, presenting the following challenges:

**Sensitivity to Occlusion and Visibility.**   To accurately compute gradients for material estimation, the object's deformation must be clearly visible. In many "in-the-wild" interaction videos, the human hand or body severely occludes the object during critical deformation phases (e.g., kneading dough). Such occlusion disrupts the differentiable rendering loss (Equation 4), preventing the model from matching simulated pixels to the ground truth. Consequently, we rigorously curated scenes in which interactions induce deformation while maintaining object visibility, thereby precluding the use of the vast majority of online videos.

**Sensitivity to Motion Blur and Speed.**   Differentiable physics demands precise temporal alignment. Fast interactions (e.g., rapid shaking or bouncing) often introduce significant motion blur and rolling shutter artifacts in monocular videos. This blur degrades the structural fidelity of 3DGS and impairs tracking initialization (Section 3.2). The resulting noise propagates into the physics simulation, causing the optimization to diverge or yield incorrect stiffness estimates. Thus, we restricted our dataset to dynamic motions captured with sufficient clarity to ensure simulation stability.

**Geometric Topology and Continuum Assumption.**   Our method employs MPM with a hyperelastic constitutive model, which assumes a continuum volume. Constructing a stable "internal filling" (Section 3.5) for objects with complex topologies (e.g., thin, knotted wires or porous sponges) is mathematically non-trivial. Inaccurate internal volume approximation leads to simulation instability. We prioritized objects with well-defined geometries (e.g., ropes, caps, toys) to ensure the evaluation focuses on the physics simulation itself rather than meshing artifacts.

**Distribution of Interaction Modalities.**   We acknowledge the uneven distribution of interaction types in our dataset, with "Point Adherent" cases predominating. This distribution is deliberate and driven by three factors: (i) **Natural Long-Tail Distribution:** In real-world manipulation tasks, sustained grasping or holding (Point Adherent) constitutes the vast majority of interactions compared to transient actions like poking or throwing. This pattern aligns with large-scale human-object interaction datasets Chao et al. (2021), where stable grasping is the primary interaction mode. (ii) **Complexity of Instantaneous Contact:** "Surface Non-Adherent" interactions (e.g., collisions) are typically instantaneous. The post-collision dynamics are highly chaotic and sensitive to microscopic geometric variations. Reconstructing such dynamics from coarse monocular video is often an ill-posed problem compared to the sustained deformation observed in adherent cases. (iii) **Temporal Resolution Constraints:** "Point Non-Adherent" interactions often occur at high speeds, leading to the motion blur issues discussed above, which degrade the reliability of physical estimation.

To conclude, our dataset should be viewed as a controlled benchmark for validating the feasibility of visual-to-physical parameter estimationa "precision-first" approach. This trade-off between dataset scale and physical fidelity is consistent with concurrent works in physics-based vision Jiang et al. (2025), which similarly utilize focused, high-quality datasets to ensure valid simulation convergence. Scaling this to "in-the-wild" datasets requires future breakthroughs in occlusion-robust differentiable rendering and more flexible constitutive models.

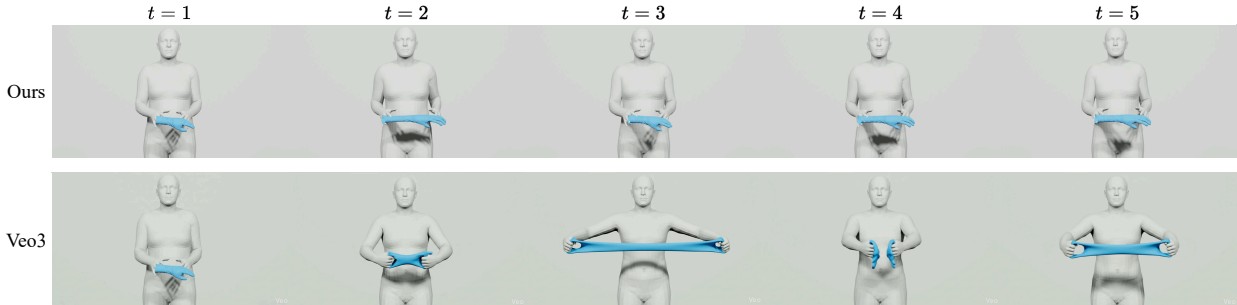

Figure 7: Comparison with state-of-the-art video diffusion models (Veo3). We conditioned Veo3 with the initial frame and a detailed prompt explicitly requesting "elastic deformation" without tearing. However, as highlighted in the Veo3 column, the model suffers from physical hallucinations: the rubber glove fractures and disconnects during stretching ($t = 2$, $t = 3$), and its topology is inconsistent. In contrast, our method uses a physics-based simulation based on estimated material properties, ensuring the object maintains structural integrity and exhibits realistic elastic behavior consistent with the physical world.

## A.2 Comparison with Video Diffusion Models

To address the capabilities of large-scale, closed-source video generation models, we conducted a qualitative comparison with Google's Veo3, a state-of-the-art video diffusion model. While such models demonstrate impressive capabilities in open-domain video generation, we investigate their limitations in scenarios requiring strict physical plausibility and material consistency.

**Setup.** We perform an image-to-video generation task using the initial frame of our "Rubber Glove" sequence as the conditioning input for Veo3. To ensure the model has the best possible context, we engineered a detailed text prompt describing not just the visual appearance, but the specific dynamics and material properties:

> *"A person holding a rubber glove with both hands, camera position fixed and static. The person performs a stretching motion by pulling the glove outward with both hands, then releasing it back to normal position. This stretching and releasing action repeats three times in a smooth, rhythmic cycle - hands pull apart, hands come together, hands pull apart, hands come together, hands pull apart, hands come together. The rubber glove stretches and elongates when pulled but never tears or breaks, showing elastic deformation. The glove returns to its original shape between each stretch. The motion is continuous and fluid, emphasizing the elasticity and durability of the material. No camera movement, stable framing throughout."*

Despite this detailed instruction, the comparison (visualized in Figure 7) highlights two critical limitations of purely generative approaches: (i) **The Ambiguity of Text-Driven Dynamics.** Natural language is inherently limited in describing precise physical interactions. While terms like "stretching" or "elastic" convey a semantic concept, they do not quantify the underlying mechanics (e.g., the specific Young's modulus or the stress-strain relationship). As observed in the Veo3 results, the model struggles to infer the correct resistance and deformation range solely from text, leading to motions that are semantically related but physically inaccurate. (ii) **Physical Hallucination and Topological Inconsistency.** A more severe issue is the lack of physical grounding, leading to hallucinations. As shown in Figure 7 (Veo3 column), during the stretching phase, the diffusion model generates a topology violation where the glove fractures into disconnected components and then inexplicably fuses back together. This "breaking and healing" artifact occurs because video diffusion models operate in pixel space based on learned statistical correlations, rather than ensuring material constancy or mass conservation. In contrast, PhysInteract (Ours) explicitly models the object as a continuum material governed by the laws of physics (MPM). By solving for the physical parameters (e.g., stiffness) rather than predicting pixel values, our method guarantees that the generated

Table 5: Runtime Comparison. We report the approximate runtime per scene for PhysInteract and baseline methods. Optimization-based physics methods (Ours, PhysDreamer) incur higher computational costs due to their differentiable simulation loop, which is a trade-off for achieving higher physical fidelity than kinematic or zero-shot approaches.

| Method | Paradigm | Runtime (min) | Hardware |
|---|---|---|---|
| PhysInteract (Ours) | Physics-based Optimization | $\sim 30$ | NVIDIA A40 |
| PhysDreamer | Physics-based Optimization | $\sim 25$ | NVIDIA A40 |
| DreamGaussian4D | Kinematic Optimization | $\sim 10$ | NVIDIA A40 |
| GPT+PhysGaussian | Zero-shot Inference | $\leq 1$ | NVIDIA A40 |

dynamics are not only visually plausible but also physically valid, maintaining structural integrity and material consistency throughout the interaction.

### A.3  Computational Cost and Runtime Analysis

To assess the efficiency of PhysInteract, we compare its runtime with that of baseline methods. All experiments were conducted on a single NVIDIA A40 GPU. Runtime performance is categorized by method paradigm: physics-based optimization (PhysInteract, PhysDreamer), kinematic optimization (DreamGaussian4D), and zero-shot inference (GPT + PhysGaussian). As shown in Table 5, PhysInteract requires approximately 30 minutes per scene. This cost is dominated by the differentiable MPM simulation loop, which is necessary to ensure physical plausibility. This runtime is comparable to PhysDreamer, which also employs an optimization-based physics approach. In contrast, DreamGaussian4D is faster ($\sim 10$ mins) as it optimizes surface deformation without solving governing physical equations, but at the cost of physical accuracy. GPT + PhysGaussian operates as a feed-forward method; while its inference is rapid, it lacks the ability to refine parameters based on video evidence, often resulting in lower fidelity (Table 2).

### A.4  User Study Protocols

To ensure a fair and unbiased evaluation of perceptual quality, we conducted a user study involving 50 participants. The participants were recruited from students enrolled in the computer vision course and consisted of graduate students and researchers with general knowledge of computer graphics and computer vision, but who were not authors of this paper and were blinded to the specific methods being tested.

**Experiment Design.** The study used a paired-comparison (A/B testing) and rating format. For each of the 11 scenes in our dataset, participants were shown the Ground Truth video alongside the results from PhysInteract and the three baseline methods (PhysDreamer, DreamGaussian4D, GPT+PhysGaussian). All methods were anonymized (labeled as "Method A", "Method B", etc.) and their display order was randomized for every trial to prevent order bias.

**Questionnaire.** For each scene, participants were asked to answer two specific questions using a 5-point Likert scale (1 = Poor/Unrealistic, 5 = Excellent/Realistic):

- **Motion Realism (MR)**: "How physically plausible is the object's motion? Consider factors like weight, elasticity, and response to interaction. Does the object move as you would expect in the real world?"

- **Visual Quality (VQ)**: "How high is the visual quality of the generated video? Consider factors like texture details, structural integrity, and the absence of artifacts or noise."

We collected a total of $50 \times 11 = 550$ ratings per method. The results reported in Table 4 represent the mean score across all participants.

### A.5 Potential Applications

We envision that PhysInteract can facilitate the following applications:

**Interactive Virtual Asset Creation.** Unlike traditional 3D reconstruction methods (e.g., 3DGS or NeRF) that capture only static geometry or kinematic motion, PhysInteract recovers intrinsic material properties (e.g., Youngs modulus). This allows reconstructed assets to come "alive" in virtual environments. In VR/AR/Metaverse contexts, users can interact with these scanned objects (e.g., poking a pillow, pulling a rubber toy) and receive physically plausible feedback, rather than merely replaying recorded animations.

**Simulation-Ready Digital Twins for Robotics.** PhysInteract serves as a pipeline for "Sim-to-Real" transfer. By observing human interactions in videos, robots can infer the stiffness and deformability of objects (e.g., distinguishing between a soft fruit and a hard ball) prior to physical contact. This visual-to-physical inference capability is crucial for robotic manipulation tasks that require precise force control to handle delicate objects without damage.

**VFX and Animation Automation.** Current VFX pipelines often require artists to manually tune simulation parameters to match a reference video. PhysInteract automates this process by inversely estimating the physical parameters that produce the observed motion. This enables the automatic generation of realistic background object dynamics (e.g., a swaying curtain or swinging rope) from reference footage, significantly reducing manual labor in 3D animation production.

**E-commerce and Virtual Try-On.** In online retail, visualizing material behavior is as critical as appearance. PhysInteract could be extended to model fabric stiffness and drape from product videos, allowing customers to observe how a garment deforms or stretches in motion, offering a dynamic experience beyond static image overlays.

