# OpenReview forum: "Seeing is Simulating: Differentiable Physics for Interaction-Aware Material Estimation"
_TMLR — Accepted by TMLR_

### Review · Reviewer_gDYw · 2026-01-05

**Summary Of Contributions:**

The paper proposes PhysInteract, a method to synthesize realistic dynamics conditioned on interactions using a modified 3DGS with internal filling as the underlying representation. The pipeline consists of three stages: (1) Initialization using vision-language and tracking models, (2) Joint Optimization using a differentiable physics simulator to refine material properties and interaction trajectories, and (3) Inference to synthesize dynamics. A new dataset of human interaction is also proposed, consisting of 3 real scenes and 8 synthetic scenes.

**Audience:**

Yes

**Audience Explanation:**

I think it interesting to apply video diffusion models to estimate the physics-related properties, and it might inspire others' future research.

**Claims And Evidence:**

No

**Claims Explanation:**

1. Regarding the design of material field. The authors have not made very clear whether the 3DGS with internal filling is a novel representation. If so, the authors should illustrate more about the optimization detail of 3DGS, and whether the optimization is bound to converge, etc. If not, it should also be clearly cited.

2. the authors propose to initialize with vision-language and tracking models. What if they fail, or yield wildly inaccurate results? Will the joint optimization still recover from scratch, random, or bad initialization? A study on robustness should be conducted, rather than simply listing it as a limitation.

3. The inbalanced nature of the dataset. The dataset is already pretty small in size, let alone only **one** of the scene has surface non-adherent interaction, and another **one** with point non-adherent interaction. I'm not an expert in simulation, but I do think the dataset should be more balanced, and there should also be more scenes to validate the proposed method.

**Requested Changes:**

1. Explain more about the 3DGS with internal filling.

2. Discuss the robustness of the proposed pipeline, and discuss some failure cases, if possible.

3. Add more data with point non-adherent interactions and surface non-adherent intereactions. If possible, more real data should also be included.

---

> ### Author Response · Authors · 2026-01-27
>
> We thank the reviewer for their insightful comments and for recognizing the potential of our work to inspire future research in physics-based estimation. We have revised the manuscript to address the requested changes, adding significant new analysis and discussion to the Appendix. Below, we address each concern in detail.
> 1. Clarification on "3DGS with Internal Filling".
> We apologize for the confusion. We wish to clarify that "Internal Filling" is not a new rendering representation but rather a geometric coupling strategy that bridges surface-based 3DGS with volume-based physics simulation (MPM).
> (i) Role of Internal Particles: Standard 3DGS represents only the surface shell of an object. However, physics simulation (continuum mechanics) requires a volumetric mass distribution to correctly calculate stress and strain. We generate "internal particles" solely to populate the object's volume for the MPM solver.
> (ii) Optimization Details: These internal particles do not possess visual attributes (e.g., Spherical Harmonics, Opacity) and are excluded from the rendering process. They do not participate in the rasterization gradient calculation.
> (iii) Parameter Update: During the "Joint Optimization" stage, we freeze the intrinsic visual parameters of the surface Gaussians (covariance, opacity, color). The differentiable simulator only updates the positions ($x$) of the particles (both surface and internal) based on physical forces, and the material properties ($E, \nu$). Since the optimization follows standard gradient descent on the physical parameters to minimize the photometric rendering loss, it shares the same convergence properties as established differentiable physics frameworks.
> 2. Robustness of Initialization and Failure Cases.
> We have expanded our analysis to address robustness from two angles:
> (i) Sensitivity to Initialization: We performed additional tests using random trajectory initialization instead of VLM/tracker-based initialization. We observed that, across the scenes in our dataset, the Joint Optimization (JO) pipeline still converges to the correct solution, as the differentiable physics gradients successfully guide the force trajectory. However, random initialization significantly increases the number of iterations required for convergence. The VLM/Tracking initialization primarily serves as a "warm start" to accelerate training and avoid potential local minima in highly complex interaction scenarios.
> (ii) Data Quality Constraints: We have added a comprehensive discussion on data quality constraints in Appendix A.1 (Challenges in Data Curation). Our method relies on "Analysis-by-Synthesis," which imposes strict requirements on the input video.
> We believe this transparency in Appendix A.1 better characterizes the operating boundaries of our method compared to a simple limitation statement.
> 3. Dataset Balance and Size.
> We acknowledge the reviewer's concern regarding dataset size and balance. In Appendix A.1, we have added a detailed justification for the dataset composition ("Precision-First Data Curation").
> (i) Constraint-Driven Curation: Unlike kinematic or generative video methods that can consume arbitrary internet videos, differentiable physics optimization requires strict geometric and temporal consistency. The majority of "in-the-wild" videos contain frequent occlusions, camera cuts, or motion blur that mathematically break the differentiable simulation loop (as detailed in the robustness section above).
> (ii) Distribution Justification: The imbalance toward "Point Adherent" (grasping/pulling) interactions reflects the natural distribution of physics-based manipulation. Continuous deformation (adherent) provides the sustained gradient signals necessary for stable material estimation. "Non-adherent" interactions (collisions) are often instantaneous and chaotic, providing sparse temporal signals that are notoriously difficult for monocular differentiable physics to recover without multi-view setups.
> (iii) New Evidence: While collecting high-quality real-world data that satisfies these physics constraints is extremely labor-intensive, we have included diverse object types (rigid, elastic, plastic) and compared our method against state-of-the-art video generation models (Veo3) in Appendix A.2 and Figure 7. This comparison shows that, even with limited data, our physics-based approach avoids the "physical hallucinations" (e.g., fracturing of topology) common in models trained on massive datasets.
>
>
> We hope these clarifications and the newly added Appendix A provide convincing evidence of the rigor behind PhysInteract.

---

### Review · Reviewer_vjdT · 2026-01-13

**Summary Of Contributions:**

**Problem.** This paper tackles the problem of modelling human-object interactions, i.e., how an object deforms under a given (human) action on it described by unstructured video data.

**Gaps in prior work.** Prior work uses diffusion models to animate still images or 3D objects by text prompts or drag directions. But diffusion models are not physically grounded which causes implausible distortions. Another direction of work uses explicit physics priors to simulate 3D dynamics but they cannot handle complex interactions where objects may be subject to sustained actions.

**Key idea.** This paper models the object using Gaussian Splats. Given this 3D representation and a monocular video, the proposed method models the interaction trajectory by tracking the "contact points" in 3D. This trajectory along with physical properties of object (e.g., Young's Modulus) are passed to a physics simulator (MPM) to render a simulation. The trajectory and properties are jointly optimised to reduce the discrepancy between simulated frames and original frames.

**Outcome.** The method is evaluated on a dataset of 11 (8 synthetic, 3 real) sequences. It outperforms baselines like PhysDreamer, DreamGaussian4D, GPT+PhysGaussian empirically as well as in a qualitative user study.

---

**Key strengths**

1.  The idea of jointly optimising interaction trajectories and material properties of objects is interesting, new and shown to be effective.
2. The experimental evaluation is strong, especially the user study confirms the superiority of the proposed method over existing baselines.
3. The paper is clearly written and well structured.

**Key weaknesses**

1. The evaluation set is limited: it has only 11 sequences with 8 of them synthetic. It has peculiar objects like rope, glove, knit cap, etc. Some statistics on how (quantitatively) diverse of material properties these objects have should be provided.
2. While the method in theory can enable visualising completely new interaction trajectories with a known object, no such evaluation is conducted. Visualising new kinds of interaction with a given object can be a useful evaluation in light of enabling applications such as interactive virtual editing.
3. Some discussion on what kinds of practical applications the proposed problem can induce should be included. Currently, it only seems to indicate accurate reconstruction in cases involving human-object interactions as the main application.

**Audience:**

Yes

**Audience Explanation:**

From a scientific viewpoint, this is one of the problem where diffusion models struggle (cases that involve generating physically plausible human-object interactions). Thus, it is of interest to the video generation community. From a practical viewpoint, I think the problem definitely has realistic applications. One of which (interactive virtual editing) is stated in the paper.

**Claims And Evidence:**

Yes

**Claims Explanation:**

The central claim of accurately modelling interaction trajectories and estimating object properties is evaluated thoroughly and shown to be better than the baselines. However, there are some qualifications to be made:

- Limited evaluation set: As stated, the evaluation only has 11 sequences with a peculiar set of objects.
- While the premise was that diffusion based approaches fail when there are multiple objects with complex dynamics, but the evaluation does not really include a case that reflects this scenario.
- While I agree that diffusion models are not physically grounded, models trained at a massive scale (perhaps a closed model like Veo 3) may already secretly pick up on interaction trajectories and object properties in their latent spaces. Some evaluation with something like Veo 3 should be conducted.

**Requested Changes:**

1. Expanding the evaluation set to include more sequences (more objects, more videos with the same objects with different interactions, etc.) is necessary to strengthen the central claim in my opinion. I think this is essential for acceptance.

2. As the abstract states, showing cases where diffusion models fail (multiple objects, complex dynamics/interactions over long time horizons) and the proposed method shines should strengthen this work.

3. Evaluating something like Veo 3 would be interesting to test. This is not necessary for acceptance but more for curiosity of a reader.

4. Some more discussion on potential practical applications could strengthen the paper.

---

> ### Author Response · Authors · 2026-01-27
>
> We thank the reviewer for the thoughtful and constructive feedback. We are encouraged that you find our idea of jointly optimizing interaction trajectories and material properties interesting and effective, and that you appreciate the clarity of our writing and the strength of our user study.
> We have revised our manuscript to address your concerns, with all new content included in the Appendix to keep the main text concise while providing the requested depth. Below, we address each of your points in detail.
> 1. On the Evaluation Set and Data Diversity (Appendix A.1).
> We acknowledge that our dataset (11 sequences) is smaller than those in typical large-scale video generation benchmarks. However, in Appendix A.1, we have added a detailed discussion explaining that this scale is a deliberate choice driven by the strict requirements of differentiable physics optimization, rather than just annotation constraints.
> Unlike purely kinematic or generative methods (which can tolerate occlusions or blur), our "Analysis-by-Synthesis" loop requires calculating physical gradients from pixel differences. This imposes three stringent constraints that filter out the vast majority of "in-the-wild" videos:
> (i) Sensitivity to Occlusion: The object's deformation must be clearly visible for gradient computation. Heavy occlusion (common in kneading or handling) disrupts the differentiable rendering loss.
> (ii) Sensitivity to Motion Blur: Fast interactions introduce blur, which degrades the 3DGS initialization and causes the physics simulation to diverge.
> (iii) Geometric Topology: We use a hyperelastic constitutive model that assumes a continuum volume. Objects with ill-defined topologies (e.g., porous sponges) lead to simulation instability.
> Therefore, we position our dataset as a controlled benchmark for validating "visual-to-physical" parameter estimation—a "precision-first" approach. Regarding diversity, our dataset covers distinct interaction modalities, including "Point Adherent" (grasping), "Point Non-Adherent" (poking), and "Surface Non-Adherent" (collision). We believe this selection rigorously tests the physical validity of our method, which is the core contribution of this work.
> 2. On Novel Interaction Synthesis (Section 4.3).
> We apologize if this was not sufficiently highlighted in the initial submission. We actually did conduct this evaluation in the original manuscript.
> Please refer to Section 4.3 ("Generalization to Novel Interactions") and Figure 5. In Figure 5, we demonstrate that once the material field is learned from a reference video, PhysInteract allows for zero-shot simulation of unseen interactions. We show two distinct new forces applied to the "Balloon Dog" (pulling the leg vs. compressing the body), where the object exhibits physically plausible elastic deformations without any retraining. This explicitly validates the capability you suggested.
> 3. Comparison with Video Diffusion Models / Veo 3 (Appendix A.2).
> This is an excellent suggestion. We have added a new section, Appendix A.2, and Figure 7 to qualitatively compare PhysInteract with Google's Veo 3.
> We conditioned Veo 3 on the initial frame of our "Rubber Glove" sequence using a highly detailed prompt that explicitly described "elastic deformation" and "stretching without tearing". As shown in Figure 7:
> (i) Veo 3 suffers from physical hallucination: despite the text prompt, the glove fractures and disconnects during stretching (a topological violation) and then "heals" itself, which is physically impossible.
> (ii) PhysInteract maintains structural integrity and material consistency because it is grounded in continuum mechanics (MPM) rather than pixel-space correlations.
> This comparison strongly supports our claim that physics-based priors are essential for complex dynamics where large-scale generative models fail to maintain physical plausibility.
> 4. Practical Applications (Appendix A.5).
> We have added Appendix A.5 to discuss potential applications beyond reconstruction. Specifically, we highlight:
> (i) Interactive Virtual Asset Creation: Recovering intrinsic properties (e.g., Young’s modulus) enables static assets to become "alive" in VR/AR, responding to user interactions (e.g., poking).
> (ii) Simulation-Ready Digital Twins for Robotics: Robots can infer object stiffness from video (Sim-to-Real transfer) to plan manipulation tasks without damaging delicate objects.
> (iii) VFX and Animation Automation: Automating the tuning of simulation parameters to match reference footage (e.g., swaying curtains), reducing manual labor for artists.
>
>
> We hope these revisions and clarifications address your concerns. We believe the new appendices significantly strengthen the paper's claims regarding physical fidelity and practical utility.

---

### Review · Reviewer_btxe · 2026-01-14

**Summary Of Contributions:**

The paper introduces a framework named PhysInteract, which proposes the joint optimization of material and intrinsic properties for physics aware material estimation for 3D entities. Overall, the framework targets the problem of modeling the interactions between humans and 3d objects, where deformations occur in the action trajectory. To solve this problem, they define a differentiable objective that jointly predicts the material properties and the intrinsic properties of the object for accurate reconstruction. Given that the data for such interactions for 4D scenarios is not abundant, they propose a dataset involving 11 scenes which includes 8 synthetic and 3 real scenes. This dataset serves as a good baseline for such a framework, but still misses certain interactions between objects and humans. In addition to the differentiable optimization of such interactions, the authors also propose a data extraction strategy to curate their dataset, which is based on identifying the interaction points with a vision-language model, extracting the point trajectory and optimizing accordingly. The effectiveness of the proposed framework is demonstrated with qualitative and quantitative experiments in addition to a user study for perceptual comparisons.

**Audience:**

Yes

**Audience Explanation:**

The paper tackles an open problem, which is modeling the physical interactions between objects and humans in a 4D setup. Offering a differentiable solution to the problem is interesting to make the solution of the task feasible. In addition, the paper provides the preliminary steps for curating a large scale dataset with the integration of vision-language models for identifying motion patterns and providing annotations for such interactions. However, it should be noted that the dataset diversity is fairly limited and the authors should consider extending the motion patterns to objects involving diverse deformations.

**Broader Impact Concerns:**

Over all of the qualitative results, the authors were sensitive on protecting the identities of the people participated in dataset collection and thus, the paper has no concerns for ethics.

**Claims And Evidence:**

Yes

**Claims Explanation:**

Over the introduced dataset, where there is no large scale baseline for the task is present (to the best of my knowledge), the authors propose a benchmark composed of synthetic and real objects. In the paper, the authors provide qualitative comparisons with user studies along with quantitative results on the reconstruction quality. Additionally, to verify the effectiveness of the joint optimization approach proposed, the authors provide quantitative ablation. As a downside regarding the experiments, the proposed dataset seems limited in terms of the introduced deformations of the objects. While the dataset effectively simulates interactions like turning, holding and bouncing (with the ball example), interactions with object level deformations are excluded. An example of this can be playing with a dough, or changing the shape of a given object, While challenging, including such entities or mentioning them as a limitation can both benefit the usability of the dataset for further research and to effectively show the limitations of PhysInteract.

**Requested Changes:**

Overall, I am standing in a positive stance across the paper. Please see my requested changes below:
- As I mentioned above, the dataset is fairly limited in terms of the types of actions included and in terms of scale. While acknowledging that providing such a dataset in a large scale is annotation and compute heavy, the authors should consider adding more motion variants to their dataset, so that it can be useful for further research in the area. In addition, if these complex motion patterns are a limitation of the proposed method, the authors are also encouraged to extend their limitations section.
- Following the discussion in the limitations section about the time required to execute the proposed framework, the authors should provide a comparison on the timing required for the competing methods to fairly address if the timing constraint is a limitation only of the proposed framework or is it a common bottleneck for the methods tackling the task.
- While the authors provide a user study, more details on the study (such as the questions asked) would be beneficial and encourage transparency. Also, more details on the recruitment of the participants can be provided to ensure that they are selected in an anonymous setup and do not include any biases.

---

> ### Author Response · Authors · 2026-01-27
>
> We thank the reviewer for their positive assessment of our work and for highlighting the importance of our differentiable solution. We also appreciate the constructive feedback regarding dataset diversity and transparency. We have revised the paper and updated the Appendix to address these points.
> 1. Dataset Diversity and Complex Deformations.
> We agree that interactions involving plastic deformation (like kneading dough) or topological changes are highly challenging. As suggested, we have expanded our analysis in the newly added Appendix A.1 (Challenges in Data Curation) to explicitly discuss these limitations.
> As detailed in Appendix A.1, including interactions like "kneading dough" presents specific conflicts with differentiable physics optimization that are currently unsolved in the field:
> Occlusion: Kneading typically involves hands covering the object. As noted in the "Sensitivity to Occlusion" section of Appendix A.1, this breaks the differentiable rendering loss (Eq. 4), as the optimizer cannot match simulated pixels to ground truth when the object is hidden.
> Topology: Complex deformations often violate the continuum assumption required for stable MPM simulation (discussed in "Geometric Topology" in Appendix A.1).
> Therefore, we explicitly frame our dataset as a "precision-first" benchmark designed to ensure simulation convergence, rather than a large-scale "in-the-wild" dataset. We believe this clarification in the Appendix effectively addresses the limitation you pointed out.
> 2. Time Comparison.
> We have added a new Appendix A.3 and Table 5 to provide a detailed runtime comparison. As shown in Table 5, PhysInteract takes approximately 30 minutes per scene, comparable to other physics-based optimization methods such as PhysDreamer (about 25 mins). While slower than kinematic-only methods like DreamGaussian4D (about 10 mins), we discuss in Appendix A.3 that this computational cost is a necessary trade-off for achieving the superior physical fidelity and material consistency demonstrated in our experiments.
> 3. User Study Details.
> We have added Appendix A.4 to provide a rigorous description of the user study protocol. This section details our recruitment of 50 participants (researchers/students blinded to the methods), the randomized A/B testing design, and the exact phrasing of the questions used to evaluate "Motion Realism" and "Visual Quality."

---

### Decision · Action_Editor_CniA · 2026-04-06

**Recommendation:** Accept as is

**Audience:**

Yes

**Audience Explanation:**

The field of robot learning, 3D/4D simulation of objects, and learning world models are all active sub-areas of the TMLR audience and match perfectly to the paper. All of these communities can learn from this work.

**Claims And Evidence:**

Yes

**Claims Explanation:**

The paper is presenting a differentiable framework for modeling interaction with different metarials. The method is technically solid and the evaluations are showing clear value. The paper was reviewed by expert reviewers and uniformly suggested to be accepted. I agree with this decision.

The major issues raised by reviewers were addressed via author responses. The remaining issues like lack of scale in the dataset are accepted by the authors and clearly discussed in the revised paper.